# Phasing Maneuver Analysis from a Low Lunar Orbit to a Near Rectilinear Halo Orbit

**Giordana Bucchioni** [†]  **and Mario Innocenti** [*,†] 

Department of Information Engineering, University of Pisa, 56122 Pisa, Italy; giordana.bucchioni@ing.unipi.it
* Correspondence: mario.innocenti@unipi.it; Tel.: +39-221-7319
† These authors contributed equally to this work.

**Abstract:** The paper describes the preliminary design of a phasing trajectory in a cislunar environment, where the third body perturbation is considered non-negligible. The working framework is the one proposed by the ESA's Heracles mission in which a passive target station is in a Near Rectilinear Halo Orbit and an active vehicle must reach that orbit to start a rendezvous procedure. In this scenario the authors examine three different ways to design such phasing maneuver under the circular restricted three-body problem hypotheses: Lambert/differential correction, Hohmann/differential correction and optimization. The three approaches are compared in terms of $\Delta V$ consumption, accuracy and time of flight. The selected solution is also validated under the more accurate restricted elliptic three-body problem hypothesis.

**Keywords:** space rendezvous; NRHO; phasing



## 1. Introduction

Current and future plans to return to the Moon are considering the presence of a permanent space station in orbit around the L2 Lagrangian point of the Earth–Moon system. Some of the advantages are the capability of accessing the Moon's dark side, and a continuous communication link with Earth. The permanent station will host vehicles leaving the Moon to load material to be sent to Earth, and become a bridge for the astronauts in their return to our satellite.

For an ascending vehicle, a key phase is the automated phasing maneuver, where the spacecraft must raise towards the orbiting space station to accomplish the rendezvous. In a cislunar environment the design of this maneuver is more complex because of the non-negligible three-body dynamic environment.

Phasing trajectories are well known maneuvers that change the size of an original orbit to meet a final target orbit at a different point in time. They can be calculated according to a wide variety of specifications such as minimum $\Delta V$, specific time interval for the transfer, etc. The computation complexity of phasing is strictly related to the dynamic model used in a fashion similar to trajectory design. Several approaches exist in the literature, which depend on the dynamic model used to describe the vehicle's motion.

Due to the absence of closed form solution, the numerical propagation of the dynamics requires particular attention to the selection of the boundary conditions.

To this end, one possible method is based on the solution of the Lambert's problem. It relies primarily on Keplerian motion, and computes a suitable orbit given initial position, final position and time of flight [1]. Lambert's approach was proposed in the case of three-body problem as well, for instance reference [2] uses Hill's approximation to several mission scenarios of transfers between Earth and Lagrange points, including target halo orbits. However only numerical complex solutions are possible with a potential high fuel expenditure. Nevertheless, the use of Lambert's approach could be useful to determine a first guess departing trajectory from the assumed lunar orbit, since the Moon's gravity can be considered dominant over the Earth's gravity for altitudes up to 500 km [3].

Another approach can be found in [4], where a continuous low thrust maneuver computation is solved via optimal control, with a cost function dependent on the power consumption and time of travel. Although not strictly applicable to our problem, it provides an example of viable numerical optimization (using a shooting method and sequential quadratic programming).

Of more interest for the present work is the literature based on the application of manifolds theory [5] to phasing. The common denominator in these works being a target halo or NRHO orbit, a ballistic or indirect transfer sequence following selected manifolds, time of flight requirements, and fuel expenditure. Reference [6], for example, reviews the CR3BP dynamics, and invariant manifolds computation. Then the connection between manifolds is addressed, and the study is applied to far rendezvous and close rendezvous using intermediate halo orbits. In [7], the authors present a somewhat similar summary study, comparing, in terms of $\Delta V$, a ballistic direct transfer from LLO to NRHO with a transfer using an intermediate NRHO, which could be used also for status assessment. Although there is no theoretical basis for the selection of the type and number of intermediate orbits (which is left to the mission design phase), a reduction in $\Delta V$ is achieved, especially if there are no stringent constraints on the TOF. An interesting preliminary analysis of phasing, which includes safety considerations, is presented in [8]. Here the authors consider the problem of safe phasing, by testing the accuracy of CR3BP motion with an Ephemeris model and parametrizing the results with respect to the degree of "out-of-plane" component of the target halo orbit.

An additional application of manifolds theory to phasing can be found in [9]. In their paper, the authors refine the computation of the target NRHO using a Bi-Circular model, they analyze the results with a Montecarlo simulation, and they apply their method to an Earth to Moon transfer. Reference [10] presents a comparative transfer sequence to different NRHOs in terms of $\Delta V$ for trajectories computed using Lambert's arc and intermediate halo.

The present paper focuses on the phasing trajectory defined within ESA's Heracles mission, in particular it presents three different methods for the phasing maneuver and compares them. The scenario is never been studied, and has become particularly important in lieu of the future missions, as shown in [11]. The originality of the comparison will allow successive detailed mission design phases. The paper is organized as follows: Section 2 describes the working scenario and relevant CR3BP equations of motion; Section 3 describes three phasing methods and resulting trajectories; a validation using more accurate ER3BP dynamics is presented in Section 4; the discussion and conclusions are described in Sections 5 and 6, respectively.

## 2. Mission Scenario and Dynamic Model

This section reviews the models used, the main assumptions, and defines the specifics of the mission scenario considered for the design of a phasing trajectory.

The proposed scenario is motivated by the original Heracles mission of the European Space Agency [12]. The mission consists of an independent landing on the south pole of the Moon, autonomous collection of lunar samples and returning the samples to the Lunar Space Gateway. One of the key phases of this mission is the phasing procedure from a Low Lunar Orbit to the L2-NRHO. In this phase, the Lunar Ascent Element or chaser will execute $\Delta V$ changes to reach a specific location where the actual rendezvous will start.

### 2.1. Reference Frames

To better understand the equations developed in the paper, we consider the following reference systems used in the representation of the dynamics:

- **Inertial Reference Frame**: all inertial coordinate frames have their axes aligned with the axis as defined for the Jet Propulsion Laboratory (JPL) DE2000 Ephemeris files. The x-axis points in the direction of the vernal equinox at midday of the 1st of January 2000. The z-axis points in the direction normal to the mean equatorial plane, in the direction of the North pole, while the y-axis supplements the set to be a right-hand coordinate system. The center of the inertial frame can be located in the center of a celestial body or in the barycenter of a system of celestial bodies.
- **Synodic (Co-rotating) Reference Frame**: very suitable coordinate frames for the illustration and treatment of libration point orbits are co-rotating coordinate frames. The x-axis points from the primary body to the secondary body, the z-axis is the axis of the angular velocity of the bodies and is therefore normal to the orbital plane of the two primaries and the y-axis supplements the coordinate system to be a right-hand system, it can be centered on the secondary center or on the system's barycenter.
- **Local Vertical Local Horizon Reference Frame**: The LVLH frame is defined with respect to the primary body around which the target is orbiting. Denoting with $\boldsymbol{r}_{it}$ the target position with respect to the primary $i$, with $\left[\dot{\boldsymbol{r}}_{it}\right]_{\mathcal{M}_i}$ the target velocity as seen from the primary, and with $\boldsymbol{h}_{it} = \boldsymbol{r}_{it} \times \left[\dot{\boldsymbol{r}}_{it}\right]_{\mathcal{M}_i}$ the target *specific angular momentum* with respect to the primary, the LVLH frame unit vectors are defined and named as follows,

  - $\hat{\boldsymbol{k}} = -\boldsymbol{r}_{it}/\|\boldsymbol{r}_{it}\|$ points to the primary and is called *R-bar*;
  - $\hat{\boldsymbol{j}} = -\boldsymbol{h}_{it}/\|\boldsymbol{h}_{it}\|$, is perpendicular to the target instantaneous orbital plane and is called *H-bar*;
  - $\hat{\boldsymbol{\imath}} = \hat{\boldsymbol{j}} \times \hat{\boldsymbol{k}}$ completes the right-handed reference frame, and is called *V-bar*.

  The above definition of the LVLH frame is consistent with the one given by Fehse in its classical reference book for spacecraft rendezvous and docking [13]. With a little abuse of notation, the name of the unit vectors are maintained, as standard practice in the community.

### 2.2. Departure and Arrival Boundary Conditions

The boundary conditions for the problem refer to the chaser's departing orbit, target's orbit, and location of the insertion point for the beginning of the rendezvous phase [14].

The initial orbit is a polar circular LLO, with an altitude of 100 km. Assuming perfect two-body dynamics, the chaser speed is V = 1.634 km/s and the orbital period is T = 1.9613 h. The orbit is schematically shown in Figure 1, in the standard lunar inertial reference. The specific orbital elements are not of interest for the rest of the work. In the paper, the departing orbit is referred to the synodic frame, so there is a small error in the accuracy because of the 5 degrees rotation of the y-axis of the synodic frame itself. With this assumption, the results are coherent with what described in [11]. In addition, it is shown in [11] that the range of inclinations of the departing orbit between 85 degrees and 102 degrees does not compromise the accuracy of the overall phasing trajectory.

The target orbit is a L2 southern Near Rectilinear Halo Orbit (NRHO) shown in Figure 2. The plot on the left is the orbit's propagation using CR3BP, the one on the right shows the orbit propagated for one year using the Ephemeris model described in Table 1.

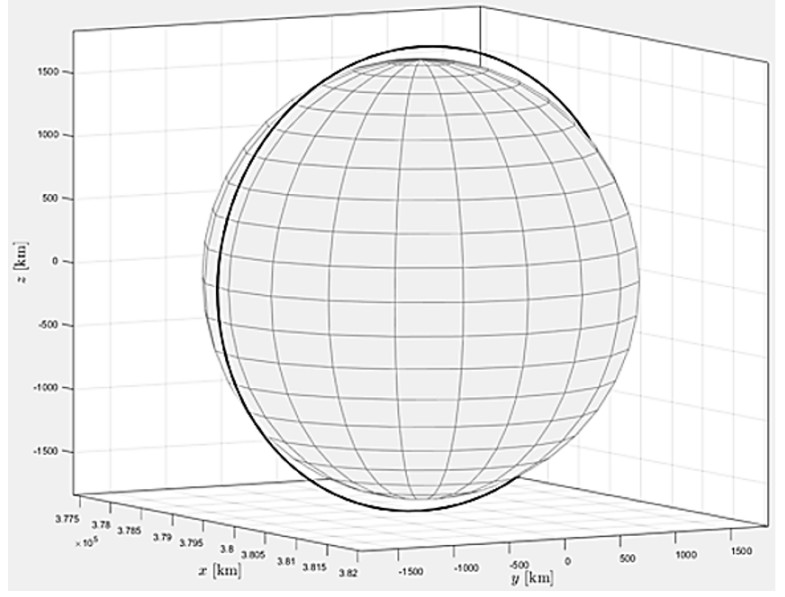

**Figure 1.** Reference low lunar orbit.

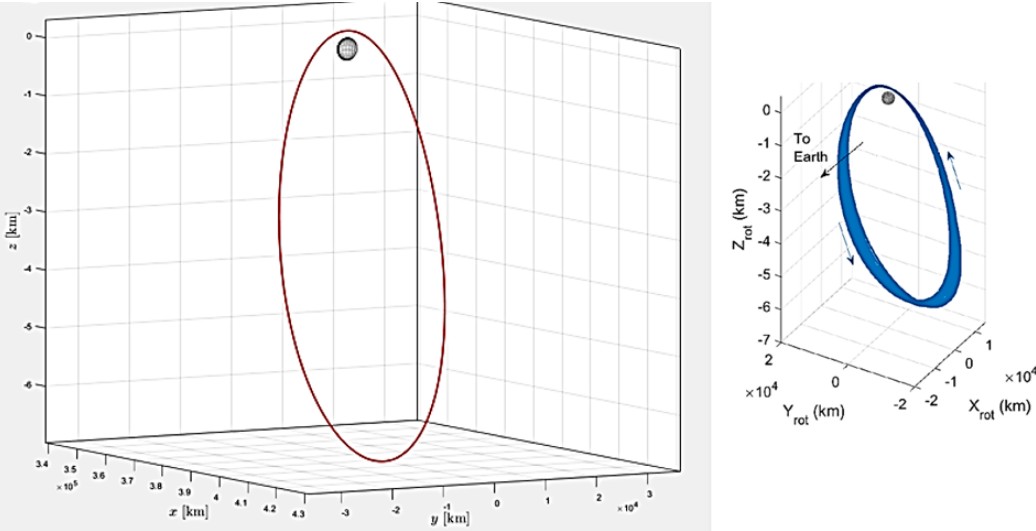

**Figure 2.** NRHO target orbit in the synodic frame.

**Table 1.** Ephemeris Data for Target NRHO Propagation ESA JD2025.

| Information | Data |
| --- | --- |
| Comment | LOP-G Southern NRHO 9:2 Resonance |
| Originator | FreeFlyer (AI-SOLUTION) |
| Object Name | LOP-G |
| Center Name | Moon |
| Start Time | 2020-01-09T00:21:00.000 |
| Stop Time | 2020-02-07T03:21:00.000 |
| Period | 6.56 days |
| Periselene | 1500 km |
| Aposelene | 70,000 km |

### 2.3. Equations of Motion

The equations of motion used to describe the phasing part of the LAE trajectory are based on the CR3BP dynamics. The gravitational field is generated by two large masses, with the primary being the Earth, and the secondary is the Moon, moving in a planar circular motion about their common center of mass. The spacecraft are considered massless. The CR3BP model is very convenient for preliminary trajectory design, and allows the computation of invariant manifolds [5,6], which can be used to drift away (unstable) or drift into (stable) specific points during phasing. Using the Synodic frame, the dynamic model is given in state space vector form by Equation (1) or in standard components by Equation (2):

$$\dot{x} = f(x(t)) \tag{1}$$

$$\ddot{x} - 2\dot{y} = -\frac{\partial \bar{U}}{\partial x}$$

$$\ddot{y} + 2\dot{x} = -\frac{\partial \bar{U}}{\partial y} \tag{2}$$

$$\ddot{z} = -\frac{\partial \bar{U}}{\partial z}$$

where the effective or augmented potential function of the system $\bar{U}(x, y, z)$ is:

$$\bar{U}(x, y, z) = -\frac{1}{2}(x^2 + y^2) + U(x, y, z) \tag{3}$$

It is possible to use a differential correction method [15] to find the desired periodic NRHO orbit, and then to compute the invariant manifolds that take the spacecraft from its initial LLO orbit to the target orbit. The technique, briefly reviewed in the Appendix A, is used to compute the best periodic approximation of the target orbit under the CR3BP hypothesis: starting from an initial guess, the initial state is constantly corrected inverting the so-called Monodromy matrix, until the propagated trajectory is a periodic orbit. The initial guess is then incrementally changed until the desired orbit is obtained.

Once the best periodic approximation of the target orbit is computed, then we use the Monodromy matrix's eigenvalues to obtain the stable and unstable manifolds. The Monodromy matrix is obtained integrating the state transition matrix over an orbital period T. The state transition matrix $\Phi$ is the matrix that linearly relates the perturbation of the initial state $\delta x(t_0)$ with the perturbation of the state after a time $t$, as shown by Equation (4).

$$\delta x(t) = \Phi(t, t_0)x(t_0) \tag{4}$$

where

$$\Phi(t, t_0) = \frac{df(x, t)}{dx}$$

The eigenvalues of the Monodromy matrix can be stable or unstable: the manifolds are naturally converging (stable) or diverging (unstable) trajectories around the periodic orbit, according to the eigenvector directions of the stable (unstable) eigenvalues. Although appealing in their genesis, they formally exist only under the CR3BP approximation.

### 2.4. Boundary Conditions

From the description at the beginning of the section, the boundary conditions are those that the chaser's trajectory should match at the ends of the phasing and they are reported in Table 2. In particular, the end of phasing, and beginning of rendezvous were provided as shown in the table. The reference orbits are shown in Figure 3. *The selected Low Lunar Orbit was used only as a starting point and no further analysis of it was performed*.

To match the desired boundary conditions, a stable manifold of the target orbit was selected around the aposelene that guarantees the convergence of the chaser towards the

target avoiding collisions at least for a whole orbit. The line in cyan in Figure 3 represents the resulting stable manifold that will take the chaser to the final desired conditions listed in the table. This is a stable manifold of the target's NRHO, which satisfies the collision-free requirements.

Figure 4 shows that the passive safety, defined from the mission's requirements (10 km), is satisfied, and the selected manifold is collision-free. Passive safety was a mission requirement, with specified values of Keep Out Zones, and approach ellipsoid around the target. Due to the highly nonlinear behavior at the Periselene, CR3BP approximation is obviously not well suited around that region.

**Table 2.** Boundary conditions for phasing.

| Initial Orbit | Insertion Point |
|---|---|
| LLO Polar | near NRHO Aposelene |
| 100 km altitude | ~50 km below (R-Bar), ~86 km behind (V-Bar) |

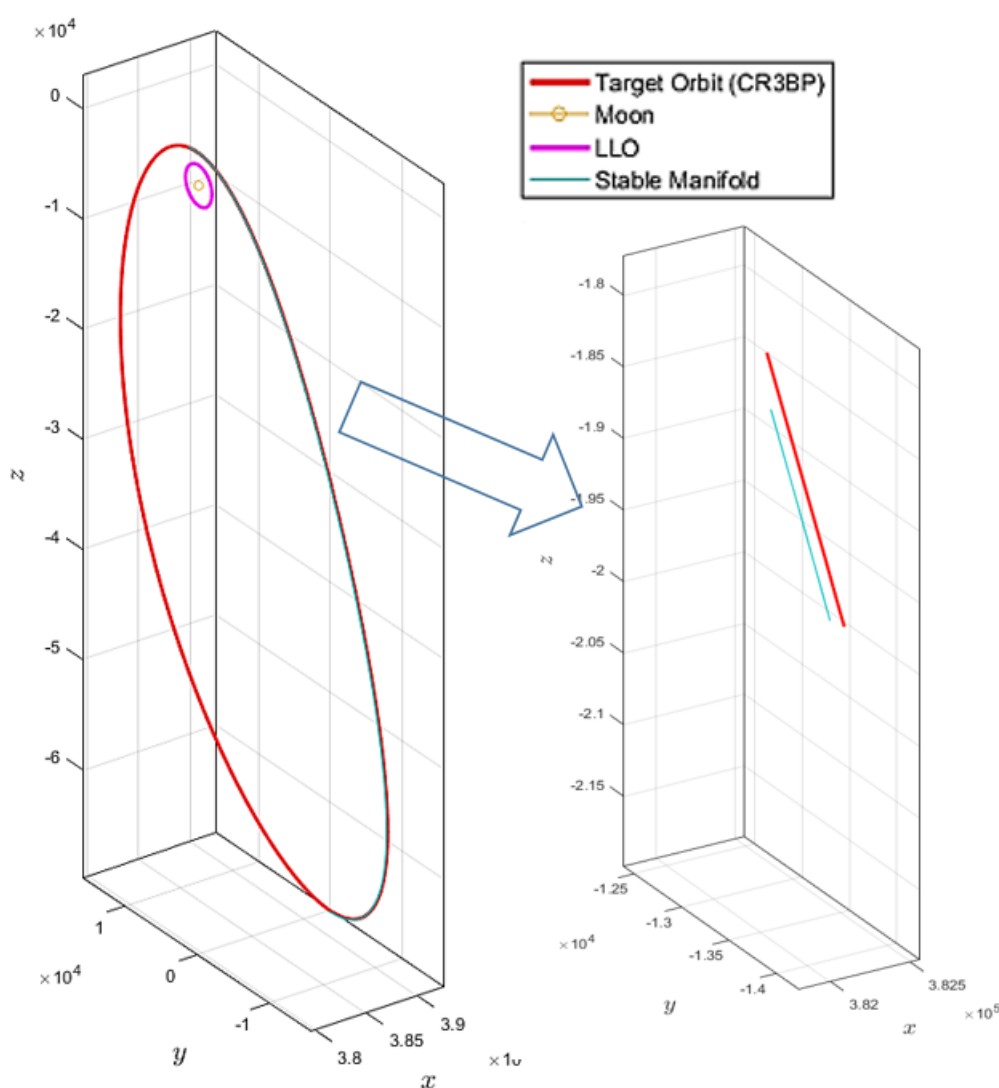

**Figure 3.** Reference orbits for phasing design.

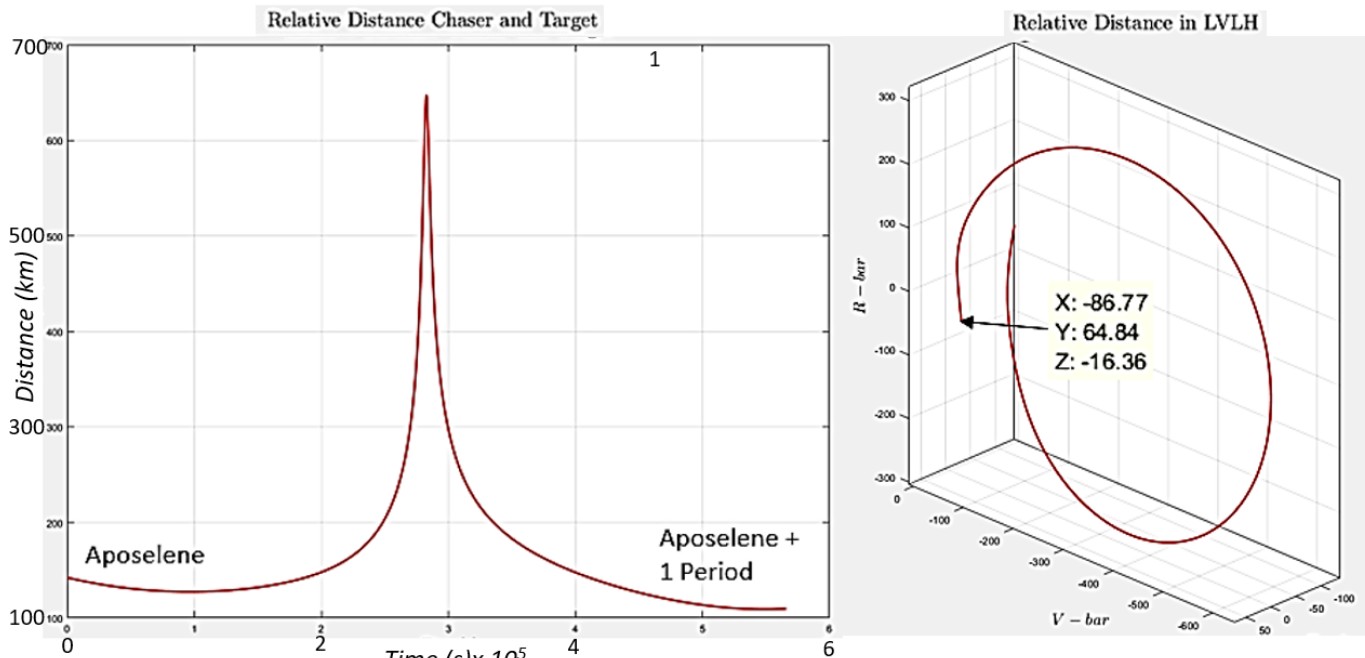

**Figure 4.** Relative distance chaser-target on the stable manifold.

### 3. Phasing Trajectory Computation

Three different methods to design the phasing maneuver are proposed in this section. The main differences are the computation of initial conditions necessary to compute the invariant manifolds. The first method is based on the Lambert's problem initial guess followed by a differential correction, the second method uses a Hohmann solution as initial guess and then the differential correction, the third method, instead, is based on an optimization approach that may lead to residual errors.

In this work we assume a two-impulse maneuver, so phasing can be computed in two steps:

- Determine a first guess on the departure velocity from the lunar orbit.
- Apply a differential correction propagation to correct the initial velocity, based on the boundary conditions at the final time, so that the spacecraft arrives at the final point with an acceptable error, along an acceptable manifold.

The initial conditions used for the propagation of the target orbit are selected as follows:

$$\begin{bmatrix} x_0 = 3.9574 \times 10^5 \text{ km} \\ \dot{x}_0 = 0 \text{ km/s} \\ y_0 = 0 \text{ km} \\ \dot{y}_0 = -0.1212 \text{ km/s} \\ z_0 = -7.1809 \times 10^4 \text{ km} \\ \dot{z}_0 = 0 \text{ km/s} \end{bmatrix}$$

The positions are then normalized to the Earth–Moon distance, and the velocities are normalized to the orbit semimajor axis times the mean motion of the CR3BP model orbit, for better numerical conditioning.

Figure 4 shows that the passive safety is satisfied, and the selected manifold is collision-free. The boundaries requested in Table 2 are indeed satisfied. The trajectory in the figure is computed with respect to the Local Vertical Local Horizon frame defined earlier.

### 3.1. Phasing Using Lambert's Initial Conditions

A classical procedure for determining an initial transfer velocity is given by the solution of the problem proposed by Lambert in the 18th century. It requires finding all

possible trajectories (or the optimal trajectory) $\mathbf{r}(t)$ that connect an initial point $\mathbf{r}(t_1)$ at time $t_1$, with a final point $\mathbf{r}(t_2)$ at time $t_2$, under a gravitational field, and with the time of flight specified by the difference $TOF = t_2 - t_1$. For the two-body case, the solution is known to provide two unique trajectories (long TOF and short TOF), which satisfy the boundaries [16,17]. In the case of CR3BP or of a gravitational field generated by three or more masses, the problem has many possible numerical solutions for any specified initial data set (such as the terminal transfer positions, transfer time, and number of complete orbits around the attracting center). This is the main reason for using Lambert's problem in the two-body setting.

The computation of a first guess on the initial velocity is crucial, and can be done using the procedure in [1], with final position and final velocity written in terms of initial position, initial velocity for a given time of flight. The required initial conditions in position and velocity can be found using the $f$ and $g$ Lagrange coefficients formulation.

The Lagrange coefficients, which are functions of true anomaly, and time, and their time derivatives are computed as in [17,18], although more recent algorithms could also be used, such as those in [10,19–21].

Once a set of initial conditions is obtained, we fix a time of flight, and the trajectory is propagated using differential correction on the linearized dynamics to reach the final state. This procedure provides the system's transition matrix. The error on final position and the inverse of the transition matrix are then used to correct the initial velocity value, and this procedure continues until the position at the final time is below the constraints specified above.

An example is shown in Figure 5. In the figure, the blue line corresponds to the trajectory evolution based on Lambert's theorem in the two-body problem. The black line is the result of the differential correction propagation using the CR3BP model. The final position corresponds to an eccentric anomaly of 180 degrees. The time of flight is obtained from Lambert's theorem application from [17], or from [1], and results in a value of 15.72 h.

**Figure 5.** Example of direct phasing using Lambert's theorem.

The plots on the left in Figure 6 shows the evolution of the trajectory propagated for different (longer) times of flight. 9 different time intervals were used from 15.72 to 78.6 h (0.655–3.275 days). Transfer trajectories propagated to the aposelene region as final point, with different TOF are shown in the same figure on the right plots. The out-of-plane nature of the transfer orbits is evident from the simulations.

**Figure 6.** Phasing trajectories: same end point, different TOF.

Figure 7 shows a summary of the tests carried out for the direct transfer using Lambert's first guess and the propagation via differential correction. The figure presents the results as function of different TOF and different final positions in the vicinity of the target orbit, combining what is described in the previous figures. The transfer trajectories were computed according to Table 3, where *T* is the target orbit period. The table shows propagation time versus the achieved final true anomaly.

**Table 3.** Time of Flight and Final Point Location.

| Propagation Time | True Anomaly |
| --- | --- |
| 1.4T | 152 deg. |
| 1.35T | 159 deg. |
| 1.3T | 163 deg. |
| 1.25T | 167 deg. |
| 1.2T | 170 deg. |
| 1.15T | 172 deg. |
| 1.1T | 175 deg. |
| 1.05T | 177 deg. |
| 1.0T | 180 deg. |

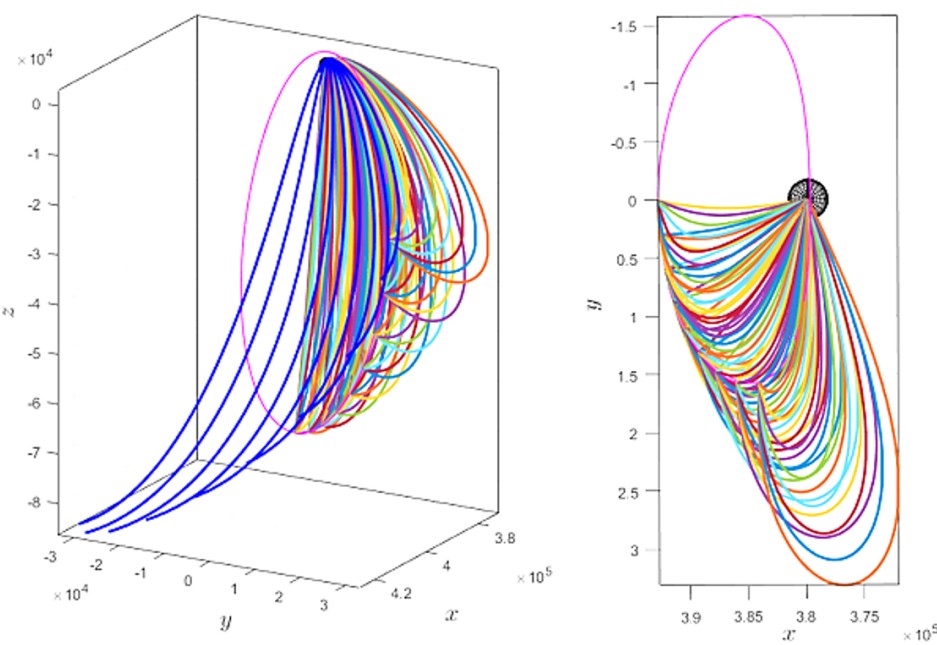

**Figure 7.** Phasing trajectories: summary results.

The computation of the $\Delta \mathbf{V}$ for the transfers shown above is performed assuming the following vector relationship:

$$\Delta \mathbf{V} = \mathbf{V}(t_0) - \mathbf{V}_{LLO} + \mathbf{V}_{manifold} - \mathbf{V}(t_0 + TOF) \tag{5}$$

The fuel expenditure depends on many factors. In this paper, we relate the $\Delta V$ to the stable manifold propagation time, and the time of flight, in order to obtain a general picture of the phasing requirements. The results are summarized in Figure 8.

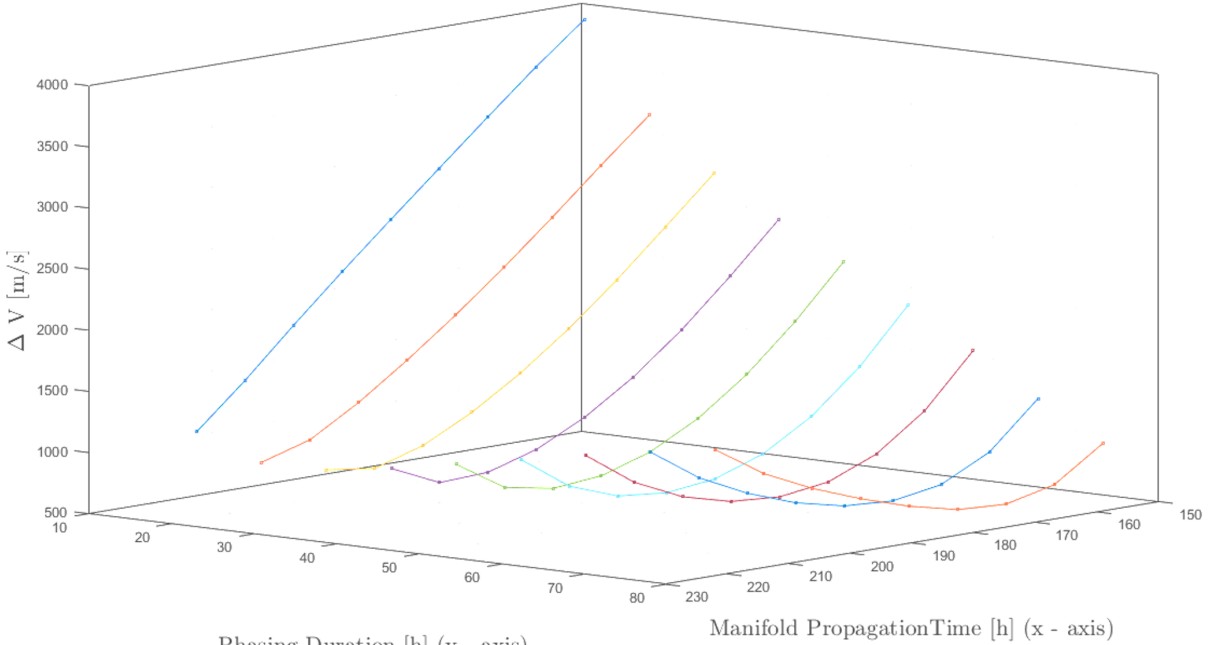

**Figure 8.** Direct phasing summary.

In the figure, the x-axis represents the propagation time (in hours) of the stable manifold from LLO to the final position location; the nine points show selected locations of the manifold starting from the aposelene. The y-axis represents the selected duration (in hours) of the phasing maneuver, and the z-axis shows the associated $\Delta V$ consumption computed according to Equation (5). A total of 81 runs were performed, in order to create the grid in the figure. The total propagation time was selected to be about 62 h.

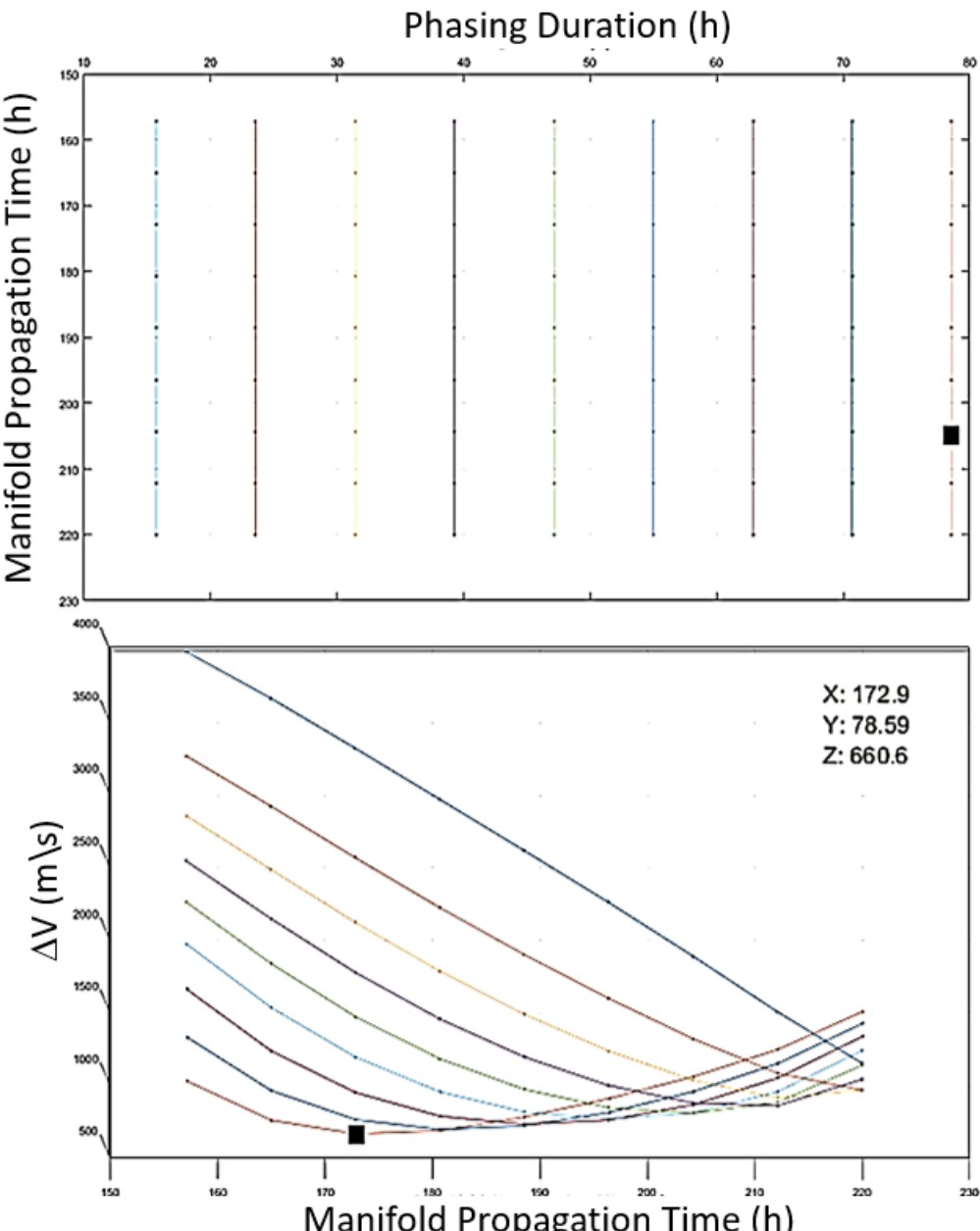

**Figure 9.** Phasing duration and $\Delta V$.

Consider, for instance, the two-dimensional projections of Figure 8 shown in Figure 9. The point in the figure indicates the amount of $\Delta V$ used for a direct phasing of 78.6 h, with a manifold propagated at time 180.8 h. From the same figure we can also determine the minimum $\Delta V$, which is about 660.6 m/s. Combining the results in Figures 7 and 8, we obtain the phasing trajectory, which produces the least expenditure in $\Delta V$. This is shown in Figure 10.

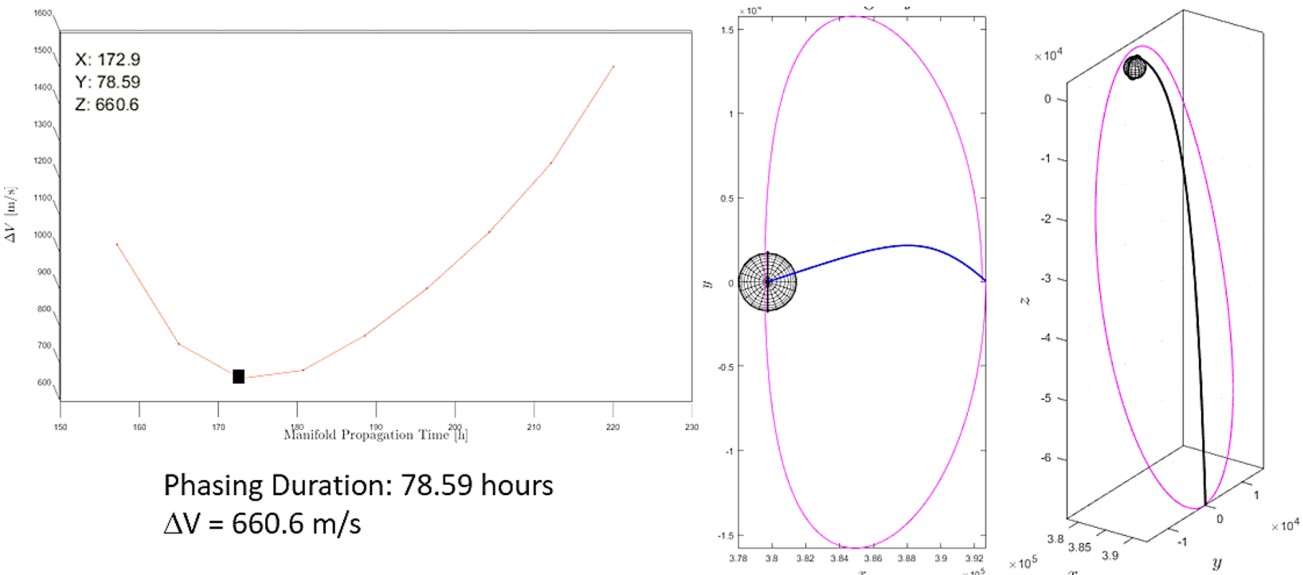

**Figure 10.** Direct phasing with Lambert: transfer orbit and fuel expenditure.

### 3.2. Phasing Using Hohmann Initial Conditions

In this case, the phasing maneuver is generated using initial position and velocity determined by a standard Hohmann transfer, then manifolds are generated as in the previous case. Using the times of flight obtained before, three trajectories were generated, and evaluated in terms of $\Delta V$. The times of flight are multiples of the orbit period (T = 6.25 days). The results are shown in Figure 11.

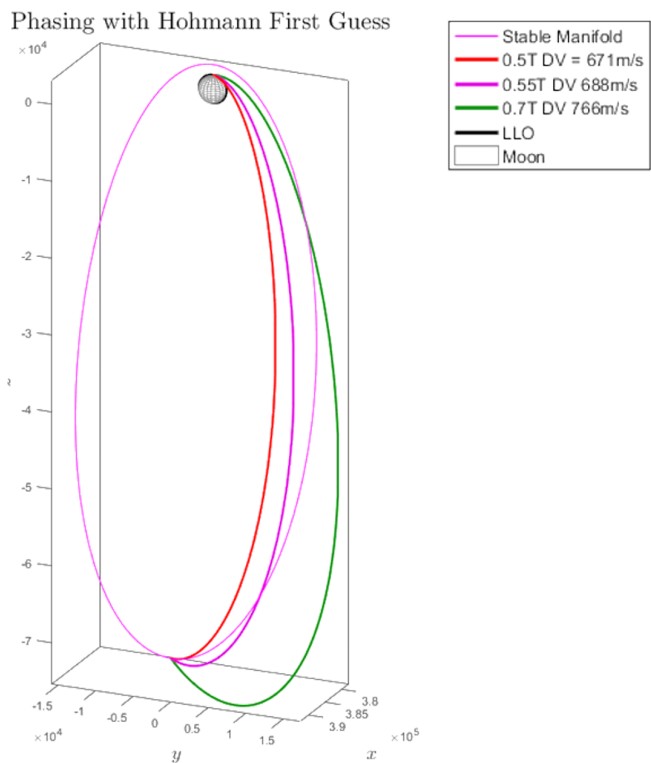

**Figure 11.** Phasing trajectories using a Hohmann as first guess.

A summary of the results is shown in Table 4. The fuel consumption is higher than the one found before, and other times of flight always produced higher energy consumption.

**Table 4.** Direct Phasing Summary using Hohmann.

| TOF | $\Delta V$ [m/s] |
|-----|-----|
| 0.5 NRHO T | 671 |
| 0.55 NRHO T | 688 |
| 0.7 NRHO T | 766 |

The results shown in this section can be qualitatively compared with those in reference [3] (Figure 5 ($\Delta V$ = 657 m/s)), and [14] Table 3.3 ($\Delta V$ = 654.3). The $\Delta V$ needed for direct transfer is similar. Potential improvement can be obtained via optimization with multiple firing, and a more precise computation of the $\Delta V$ itself.

*3.3. Numerical Optimization*

This section describes the design of a phasing sequence via numerical optimization. A phasing trajectory is designed under the CR3BP dynamic model, based on critical parameters such as position error, $\Delta V$, TOF, and number of impulses.

The main objective of this procedure is to evaluate the relative influence between final state error and fuel consumption. Therefore, a soft constraint optimization on the state was selected, which provides useful trade-off at the preliminary stage. This approach allowed information on the variation of fuel consumption as weights are changed in the cost function.

Consider the reference equations for the CR3BP model described by Equation (1) or Equation (2). The approach used is the selection of an appropriate cost function, whose minimization will produce a set of optimal (or sub-optimal) maneuvers. The optimization uses a nonlinear programming technique taken from [22]. The general problem can be formulated as that of minimizing a cost function $\mathbf{J(x)}$ with respect to the state vector $\mathbf{x}$ with constraints given by:

$$c(\mathbf{x}) \leq 0$$
$$c_{eq}(\mathbf{x}) = 0$$
$$A \cdot \mathbf{x} \leq \mathbf{b}$$
$$A_{eq} \cdot \mathbf{x} = \mathbf{b}_{eq}$$
$$lb \leq \mathbf{x} \leq ub$$

The function uses a gradient-based method that is designed to work with problems that are continuous, with at least continuous first derivatives. A sequential quadratic programming algorithm is used with updates of an estimate of the Hessian matrix at each iteration [23,24]. In our problem, we selected a quadratic cost function given by a weighted sum of the position error and the $\Delta V$. Its analytical expression is given by:

$$\mathbf{J} = q_1 ||\mathbf{e}||^2 + q_2 (\sum_i ||\Delta V_i||)^2 \tag{6}$$

The vector $\mathbf{e}$ is the error between final state and desired state,

$$\mathbf{e} = \mathbf{x}_{fin}^{Syn} - \mathbf{x}_{des}^{Syn} \tag{7}$$

and $\Delta V_i$ are the fuel consumption expenditures at each impulse. The weighting coefficients $q_1$ and $q_2$ are selectable by the designer according to the relative importance of the cost components. In the paper we report the tests obtained using the numerical values in Table 5.

**Table 5.** Weighting Values in the Cost Function.

| Weight | Value |
|---|---|
| $q_1$ | $9 \times 10^4$ |
| $q_2$ | 10 |

In the optimization, the number of impulses (thruster firings) is fixed, and simulations were performed assuming the number of impulses equal to 2, 3, 4, and 5. Spacing of the firings as well as fuel expenditure between odd/even firings were a result of the optimization and its trade-off between position error and $\Delta V$. The starting initial position was the LLO, the arrival point was the aposelene of the target orbit and the selected stable manifold was maintained the same as in the previous sections, in order to compare the errors.

The simulation results are summarized in Table 6.

**Table 6.** Fuel Expenditure as Function of the Number of Impulses (Firings).

| Number of Impulses | Total $\Delta V$ |
|---|---|
| 2 | 661 m/s |
| 3 | 709 m/s |
| 4 | 688 m/s |
| 5 | 717 m/s |

The graphical representation of the phasing trajectories in the case of 2 impulses and 4 impulses are shown in Figure 12.

The two-impulse case, although requires a smaller $\Delta V$, has a very large position vector error.

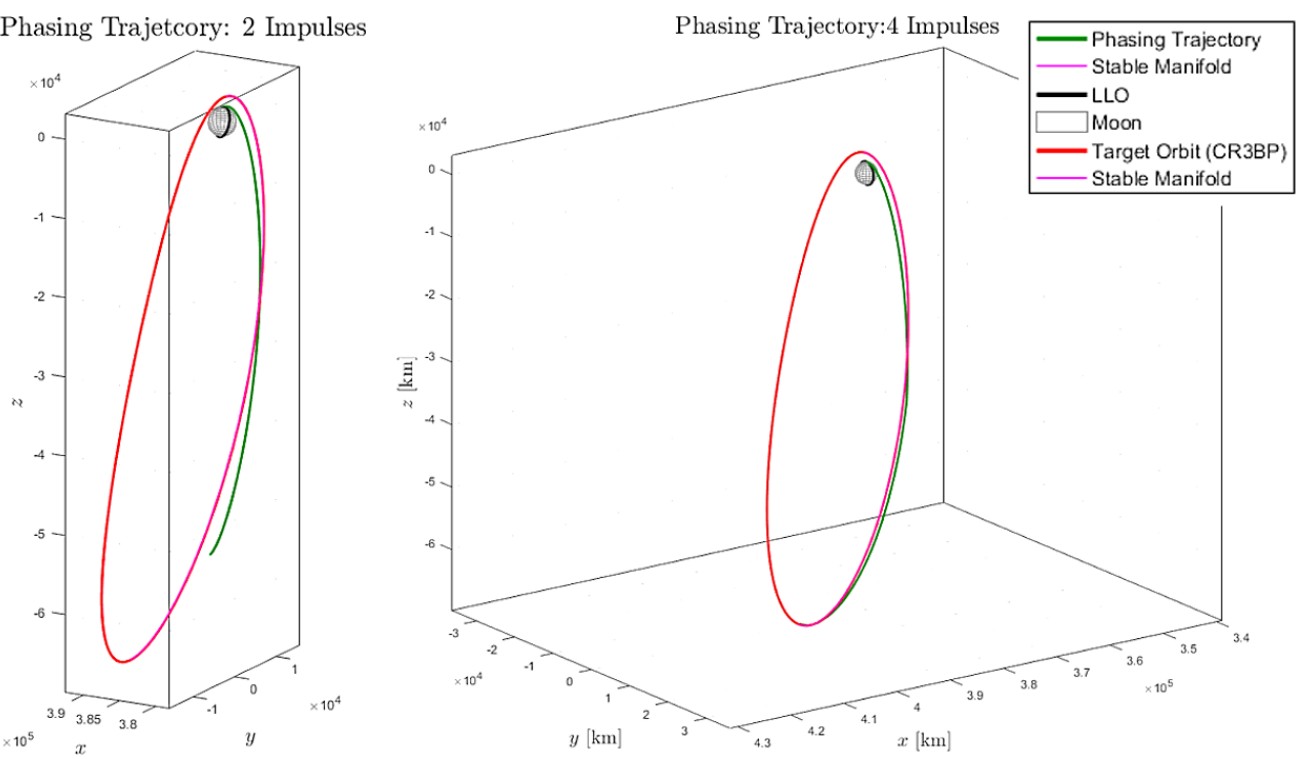

**Figure 12.** Multiple Impulse Phasing. Two- and four-impulse cases.

Based on the above results, a four-impulse maneuver was selected, since it provided a lower position error than a two-impulse case. To evaluate the influence of the selected manifold, the stable arrival manifold was then changed by varying the value of the initial perturbation in the propagation (80 km in norm) along the direction of the eigenvector associated with the eigenvalue less than 1 of the Monodromy matrix, and used for the propagation of the manifold itself. The results are shown in Table 7. Figure 13 shows the trajectory corresponding to a four-impulse with the original initial perturbation (the same as the plot on the right in Figure 12) and the one resulting from a 200 km perturbation. A change in manifold selection does not appreciatively change the fuel consumption.

**Table 7.** $\Delta V$ as Function of Manifold Selection.

| Perturbation (km) | Total $\Delta V$ |
| --- | --- |
| 80 (original) | 688 m/s |
| 90 | 687 m/s |
| 100 | 687 m/s |
| 150 | 687 m/s |
| 200 | 686 m/s |

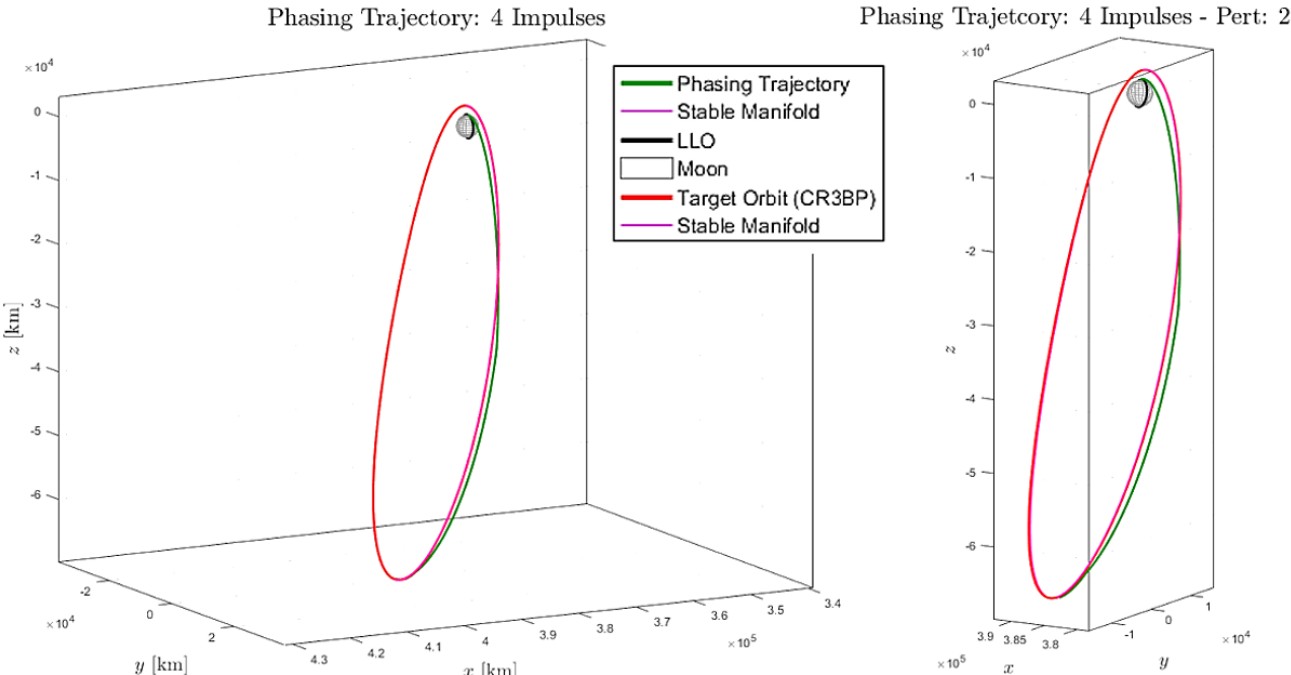

**Figure 13.** Phasing Trajectories with different Manifolds.

To verify potential $\Delta V$ reduction as a function of time of flight, the phasing maneuver was propagated for five times the target orbital period. The selected trajectory was the one corresponding to the stable manifold generated with an initial perturbation of 100 km (see third row in Table 7). The longer propagation yields an arrival point closer to the target orbit. The simulation results are summarized in Table 8. Table 9 gives the results in terms of perturbation amount.

**Table 8.** $\Delta V$ as Function of Phasing Duration.

| No. Impulses | 100 km, 5T | 100 km, 5.2T | 100 km, 5.4T |
|---|---|---|---|
| 3 | 676 m/s | 671 m/s | >700 m/s |
| 4 | **671 m/s** | 754 m/s | >700 m/s |
| 5 | 677 m/s | 1045 m/s | >700 m/s |

**Table 9.** $\Delta V$ as Function of Manifold Selection for $5 \times$ T Propagation.

| Perturbation (km) | Total $\Delta V$ |
|---|---|
| 80 (original) | 673 m/s |
| 90 | 672 m/s |
| 100 | **671 m/s** |
| 150 | 674 m/s |
| 200 | 682 m/s |

In the previous tables, the bold values indicate the occurrence of the minimum $\Delta V$. The resulting best scenario corresponds to a five-period manifold propagation, with four impulses and a total expenditure of $\Delta V$ = 671 m/s. The phasing maneuver is shown in Figure 14. The total $\Delta V$ divided into the impulse occurrence is shown in Table 10. As in all cases tested, the largest $\Delta V$ occurs at the departure from LLO.

**Table 10.** $\Delta V$ Consumption at each Impulse.

| Time of Flight $\Delta V$ | |
|---|---|
| T = 0 | 639 m/s |
| T = 1.19 day | 32 m/s |
| T = 1.28 day | 0.01 m/s |
| T = 2.2 day | 0.05 m/s |

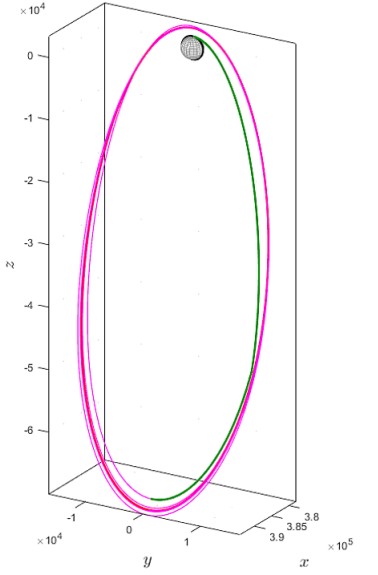

**Figure 14.** Phasing Trajectory: 32.8 days (5 times the period) manifold propagation.

If we also propagate the phasing trajectory over a five-period time, however, the distance to the manifold increases to values that do not satisfy the specifications imposed by the boundary conditions defined at the beginning in Table 2.

An example is shown in Figure 15 for a manifold computed with a 100 km initial perturbation (projection on the $x - y$ plane). The time evolution of manifold and phasing trajectories are shown on the left. On the right, we have the relative position mismatch during the orbital propagation. The increase in distance is evident already after 2 time periods. Please note that similar results were obtained for the case of 80 km perturbation.

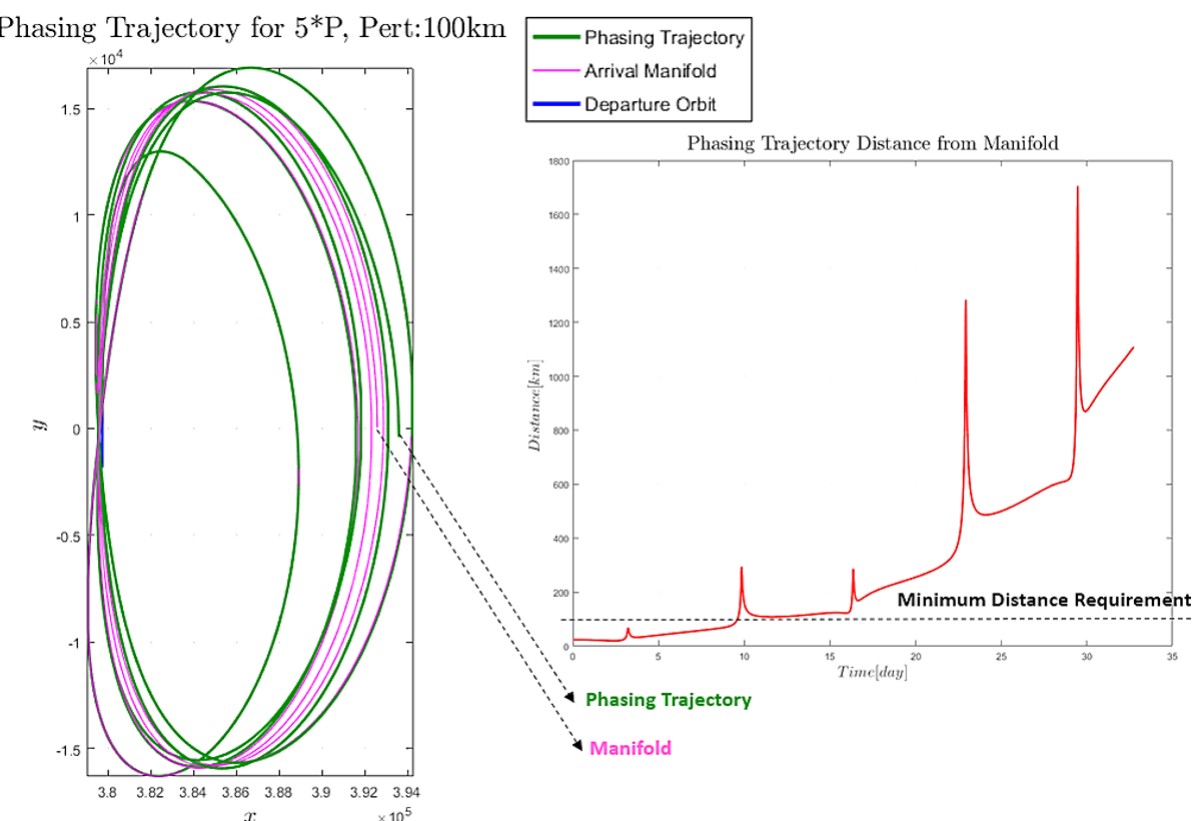

**Figure 15.** Phasing Trajectory Result, TOF = 32.8 Days.

To reduce the error between phasing and manifold, the optimization was performed by changing weighting more the error component in the cost function ($q_1 = 9 \times 10^6$). The simulation best results are shown in Figure 16 for an initial manifold perturbation of 80 km (the 100 km case is not reported here). The 80 km selection was a requirement for the initial rendezvous and proximity operations. The final position error between the phasing trajectory and the desired final position is lower than the one shown in Figure 15, at the expense of a slight increase on the total $\Delta V$.

In summary, the phasing trajectory that appears more promising is the one that reaches the target stable manifold generated with an initial perturbation of 80 km, and a propagation time of five time the target orbit period. The $\Delta V$ consumption, and the impulse timing are shown in Table 11. Even if it is obtained with a four-impulse strategy, the main $\Delta V$ is used in the first two impulses, thus comparable with the previous approaches.

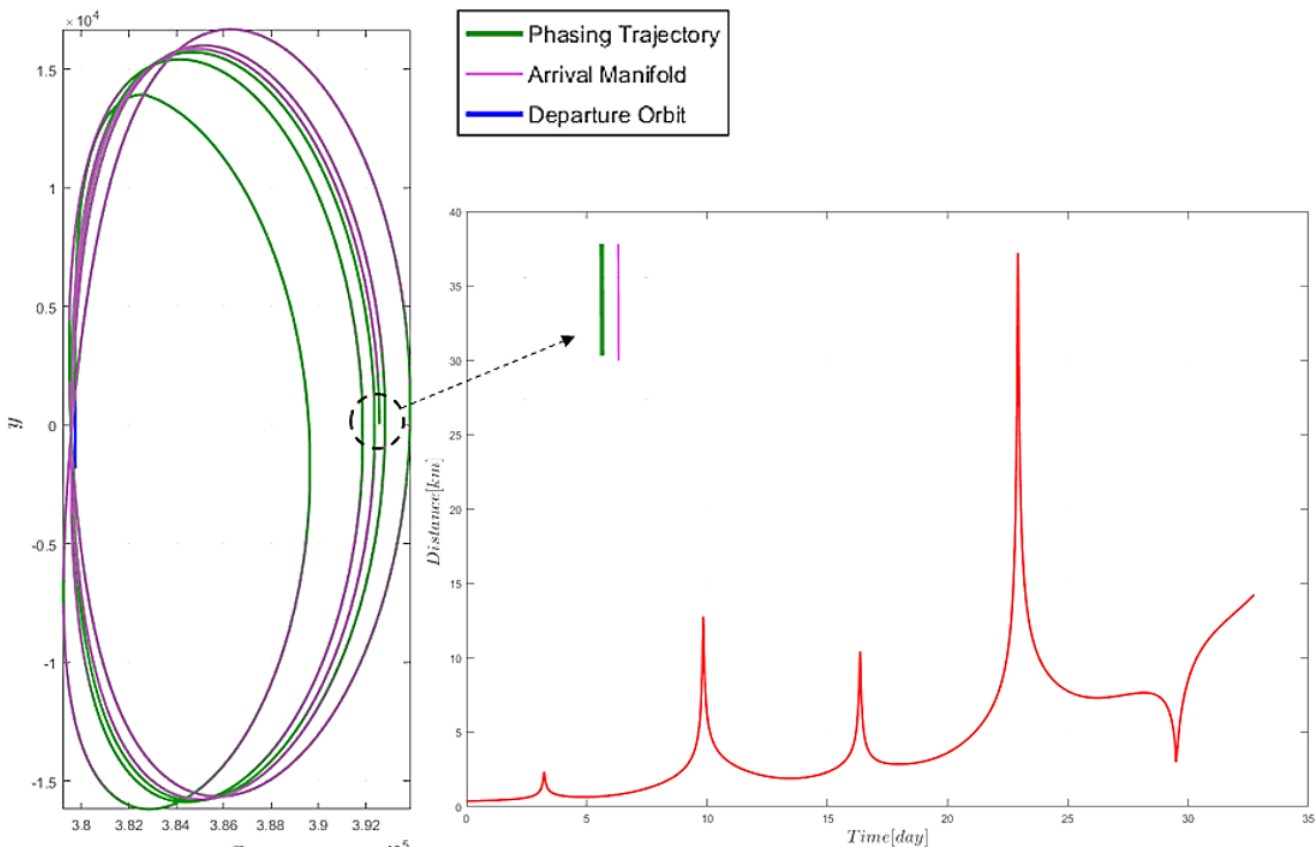

**Figure 16.** Phasing Trajectory Result, 80 km Perturbation, higher Weight on Position Error.

**Table 11.** $\Delta V$ Consumption at each Impulse.

| Time of Flight $\Delta V$ | |
|---|---|
| T = 0 | 639 m/s |
| T = 0.91 day | 39.4 m/s |
| T = 1.638 day | 0.001 m/s |
| T = 0.67 day | 0.001 m/s |

Based on the above results, some comments can be made on the results shown in Figure 16 and Table 11. First the 80 km perturbation target manifold satisfies the requirements after a time of four orbital periods, without additional maneuvers, which are necessary for a 100 km initial perturbation. This result produces the lowest position error with negligible increase in $\Delta V$.

The propagation to five orbital periods yields a phasing maneuver duration of about 701 h before satisfying the insertion requirements. However, if we remove the CR3BP restriction and perform the targeting with the real LOP-G orbit, the final position on the manifold is much closer to the orbit, than the one period case.

The real target orbit considered in the paper was obtained using the public domain "Free-Flyer" code with propagation dates 9 January 2020 to 7 February 2020. This results in a more stable phasing compared to the one using the manifold obtained with a 100 km initial perturbation.

Finally, direct targeting to the real orbit was not performed, since it would not provide any guaranteed stable drift to the target, unlike the stable manifold.

## 4. Validation Using the Elliptic Restricted Three-Body Problem

The results obtained so far use the CR3BP model, which is necessary to determine manifolds for the phasing trajectory computation. To analyze modeling errors, the equations of motion were propagated using the ER3BP model, which is one step up in the accuracy of the dynamics. The complete equation set can be found in the literature, here they are taken from [25], and summarized for clarity's sake in the Synodic frame, assuming that its origin and the origin of the Inertial system coincide.

$$\tilde{x}'' - 2\tilde{\omega}_{s/i}\tilde{y}' - \tilde{\omega}'_{s/i}\tilde{y} - \tilde{\omega}^2_{s/i}\tilde{x} = -(1-\mu)\frac{\tilde{x} - \tilde{R}_1}{\tilde{r}_1^3} - \mu\frac{\tilde{x} + \tilde{R}_2}{\tilde{r}_2^3} \tag{8}$$

$$\tilde{y}'' + 2\tilde{\omega}_{s/i}\tilde{x}' + \tilde{\omega}'_{s/i}\tilde{x} - \tilde{\omega}^2_{s/i}\tilde{y} = -(1-\mu)\frac{\tilde{y}}{\tilde{r}_1^3} - \mu\frac{\tilde{y}}{\tilde{r}_2^3} \tag{9}$$

$$\tilde{z}'' = -(1-\mu)\frac{\tilde{z}}{\tilde{r}_1^3} - \mu\frac{\tilde{z}}{\tilde{r}_2^3} \tag{10}$$

with $\tilde{(.)}$ indicating the non-dimensional value. The normalized gravitational parameter $0 < \mu < 0.5$, for the case of $M_1 > M_2$ is given by:

$$\mu = \frac{\mu_2}{\mu_1 + \mu_2} = \left(1 + \frac{M_1}{M_2}\right)^{-1} \implies \mu_2 = \mu, \quad \mu_1 = 1 - \mu$$

The objective of this computation was primarily a verification, at the simulation level, of potential divergence of the phasing over the five-period time. The simulation shows that the phasing appears possible under the elliptic motion, although the trajectory diverges after 6 orbits for the 100 km perturbation manifold, and 7 orbits for the 80 km perturbation manifold. A graphical representation for the 80 km manifold perturbation is shown in Figure 17. On the left the propagation using CR3BP, and on the right using the ER3BP where the divergence due to propagation is evident.

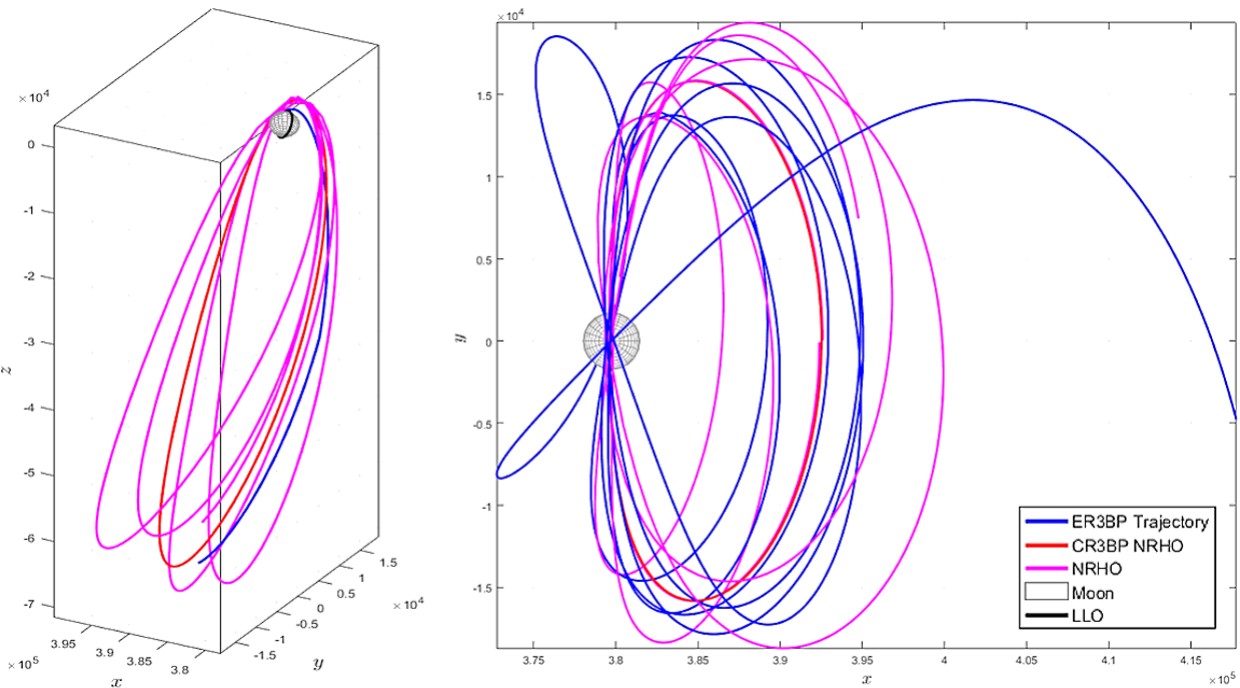

**Figure 17.** Phasing Trajectory, 80 km Perturbation, ER3BP Propagation.

Another test was to perform an optimization using the cost function in Equation (6) with weights $q_1 = 9 \times 10^6$, and $q_2 = 10$, using the ER3BP model rather than the CR3BP as reported in Figure 14, and in the third row of Table 9.

The phasing maneuver was computed for a one half of the target orbital period and for a time of flight of five times the period. The results are comparable in terms of position error; the maneuver, however, requires a total $\Delta V = 703$ m/s, as compared to $\Delta V = 671$ m/s, for the circular restricted model propagation. In addition, the optimization algorithm takes a much longer time to converge. The phasing trajectory is plotted in Figure 18, where on the left the phasing trajectory is shown for a half period TOF, and on the right for a TOF of five orbits.

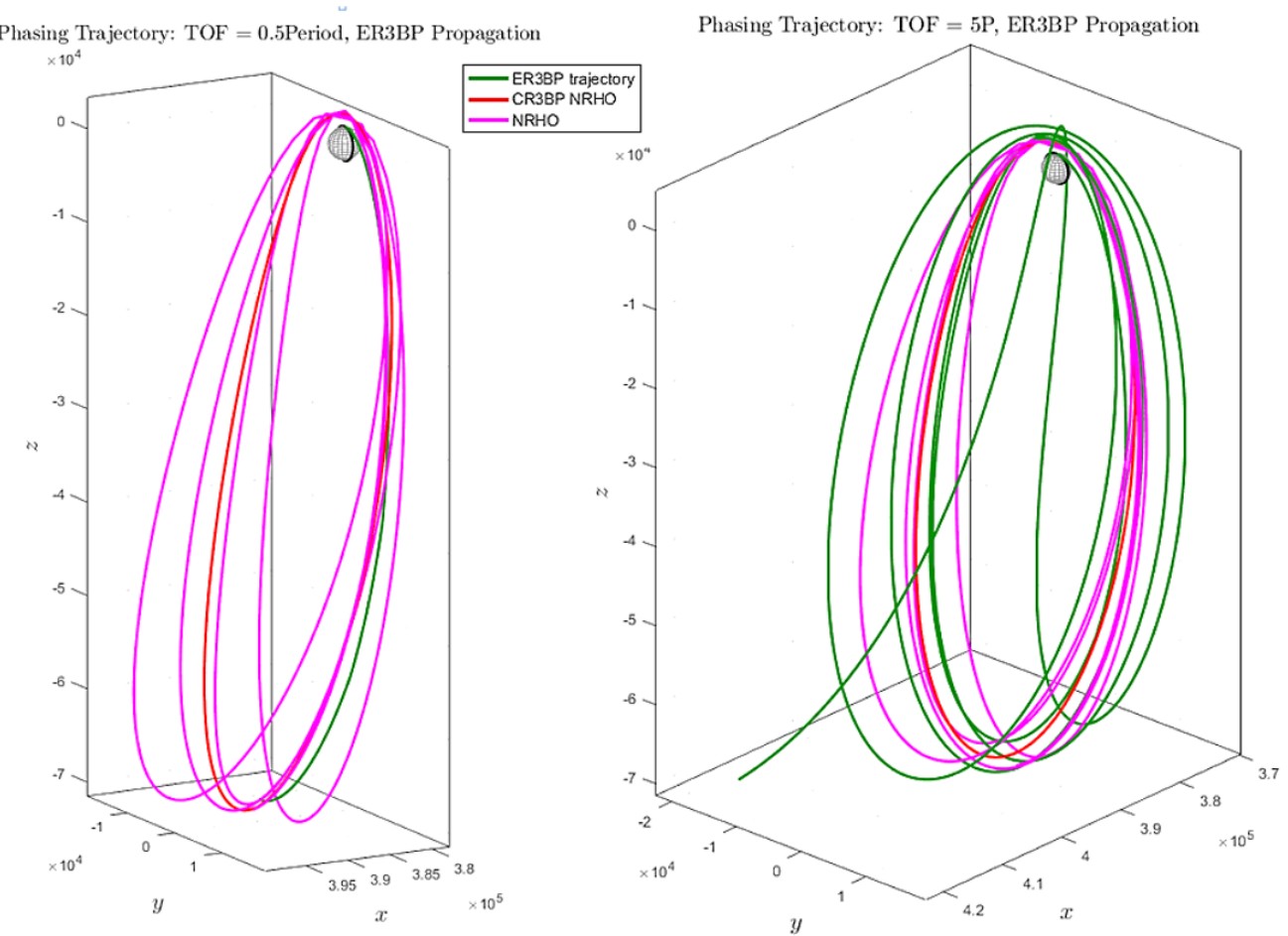

**Figure 18.** Phasing Trajectories with ER3BP Equations: 0.5T (**left**), 5T (**right**) Propagation.

*Comments on $\Delta V$ Expenditure*

All previous results have shown that the main contributor to the total $\Delta V$ is the amount required for departing LLO. This is due to the high orbital speed of the chaser, while orbiting at an altitude of 100 km from the surface of the Moon.

The situation improves if we consider LLO parking orbits at a higher altitude. For instance, when a LLO is considered at an altitude of 200 km, the total $\Delta V$ reduces to 655 m/s. This fact needs to be considered at the mission design phase, in order to optimize the total budget for the mission. Figure 19 describes a sample phasing with a time of flight of one half period , leaving LLO at 200 km of altitude. The target manifold is computed with a 100 km perturbation, and propagated for five times the orbital period.

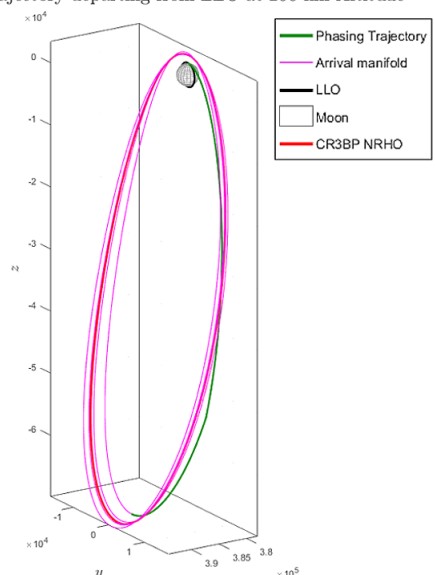

**Figure 19.** Phasing Trajectory for higher LLO Altitude.

## 5. Discussion

The study of phasing considered direct transfer from LLO to the target orbit using two-impulse maneuvers with initial conditions computed from a two-body Lambert's approach, from a direct Hohmann transfer, and a gradient-based optimization with multiple impulses. The resulting transfer trajectory, using a circular restricted three-body problem applied to Earth, Moon, was developed with the injection into a stable manifold generated by the target orbit, and selected in the direction of the eigenvector associated with the relevant eigenvalue of the appropriate Monodromy matrix.

Three approaches were studied, which defined the initial condition for trajectory propagation. A summary of the findings is reported below.

### 5.1. Lambert, Two-Impulse

- The boundary conditions needed to solve the problem are computed using Lambert's two-body problem method.
- The best simulation results are obtained with a stable manifold computed via propagation from an 80 km perturbation and it is met at the aposelene. The final position error is zero, since we are on the desired manifold.
- the time of flight of the phasing trajectory is 78.5 h (half period of the target orbit).
- The total expenditure is $\Delta V = 660$ m/s.
- The phasing trajectory is shown in Figure 10, and it is a full out-of-plane path. This is why this approach may result in an increased fuel expenditure. To be noted is the fact that the orientation of the LLO orbit was not specified, thus the amount of out-of-plane imposed by Lambert could change if different LLO orientations are chosen.

### 5.2. Hohmann, Two-Impulse

- The boundary conditions needed to solve the problem are computed using a planar two-body Hohmann approach.
- The stable manifold is computed via propagation from an 80 km perturbation and it is met at the aposelene. The final position error is zero, since we are on the desired manifold.
- The time of flight is similar to the previous case (half period of the target orbit).
- The total expenditure is $\Delta V = 671$ m/s.
- The phasing trajectory is shown in Figure 11.

### 5.3. Multiple Impulse Optimization

- The optimization procedure does not use differential correction, and it produces position errors at the final time, which depend on propagation equations, relative weight between error and fuel expenditure, optimization soft stopping conditions on the final state.
- The optimization used soft constraints on the simplified dynamics, in order to evaluate the relative influence between accuracy and fuel expenditure. Although hard constraints could have produced different results, the choice appears justified by the fact that further iterations are necessary in the mission design phase.
- A different number of impulses was evaluated, with the best results obtained for a 4-impulse sequence (see Table 5), although almost all $\Delta V$ was used with the first two impulses, making it similar to the other two solutions.
- Different stable manifolds were evaluated in terms of $\Delta V$ expenditure (see Table 7). The computed $\Delta V$ are 688 m/s and 687 m/s for a manifold propagation of one orbital period (80 km and 100 km perturbations).
- The phasing trajectory time of flight is half orbital period and it is shown in Figure 13.
- Improvement in the final position error and $\Delta V$ can be obtained by longer manifold propagation, and longer time of flight of phasing trajectory. In particular, a five-period manifold propagation, and a phasing trajectory time of flight of 701 h yields a position error within the requirements and a $\Delta V = 678.4$ m/s, as shown in Figure 16.

### 5.4. ER3BP Propagation

This limited test was performed to validate the accuracy of the circular restricted three-body model in the optimization.

- Phasing trajectory time of flight of half period, $\Delta V = 703$ m/s, see Figure 18. The figure also shows the loss of accuracy at the Periselene, as the propagation time increases.
- The validity of CR3BP is maintained only if the time of flight is much lower than the propagation time as expected.

### 6. Conclusions

The paper presents an analysis of different phasing trajectories evaluated in terms of $\Delta V$ expenditure, time of flight, and position errors achieved at the final time. Future programs of permanent return to the Moon dictated the mission scenario used in the paper. The circular restricted three-body model was found to be accurate for this study and for the assumed rendezvous around the aposelene of the target orbit. The accuracy decreases if other regions are selected for the rendezvous, and in the target orbit propagation time is extended. The initial parking orbit size is the main contributor to the total $\Delta V$ consumption and the results provide a preliminary starting point for further mission design analysis.

**Author Contributions:** Both authors have contributed equally to the work. All authors have read and agreed to the published version of the manuscript.

**Funding:** This work was partially funded by the European Space Agency, under ESA contract No. 4000121575/17/NL/CRS/hh. The views expressed in this paper can in no way be taken to reflect the official opinion of the European Space Agency. The authors would like to thank ESA officer Alexander Cropp for his valuable support.

**Institutional Review Board Statement:** Not applicable.

**Informed Consent Statement:** Not applicable.

**Data Availability Statement:** Not applicable.

**Conflicts of Interest:** The authors declare no conflict of interest.

## Abbreviations

LAE      Lunar Ascent Element
LOP-G    Lunar Orbital Platform–Gateway
NRHO     Near Rectilinear Halo Orbit
LLO      Low Lunar Orbit
TOF      Time of Flight
LVLH     Local Vertical Local Horizon
ER3BP    Elliptic Restricted Three-Body Problem
CR3BP    Circular Restricted Three-Body Problem
CLTV     Cis-Lunar Transfer Vehicle

## Appendix A. Differential Correction

The differential correction procedure is a Newton-based numerical algorithm used to improve the accuracy of trajectory estimation. Many applications can be found in the literature, in the context of the circular restricted three-body problem, for orbit determination around collinear Lagrangian points and for invariant manifold determination, for instance [26,27], among others.

The equations of motion in Equation (2) describing the CR3BP can be written in compact form:

$$\dot{\mathbf{x}} = \mathbf{f}(\mathbf{x}(t)) \tag{A1}$$

where $\mathbf{x} = [x, y, z, \dot{x}, \dot{y}, \dot{z}]^T$ is the state vector describing the motion of the vehicle. Let $\mathbf{x}(t)$ be some trajectory going from $\mathbf{x}_0$ to $\mathbf{x}_1$ under motion dynamics described by Equation (1). For any given initial condition $\mathbf{x}_0$, the resulting trajectory at time $t$ is defined by a **flow map** $\phi(t, t_0)$ as:

$$\phi(t, t_0) = \phi(t; \mathbf{x}_0) : \mathbf{x}_0 \to \mathbf{x} \tag{A2}$$

The flow map describes the state of the vehicle from its initial location to its value at the final time. Please note that the flow map satisfies the equations of motion (A1):

$$\frac{d\phi(t; \mathbf{x}_0)}{dt} = \mathbf{f}(\phi(t; \mathbf{x}_0)), \qquad with \qquad \phi(t_0; \mathbf{x}_0) = \mathbf{x}_0$$

When the initial condition is slightly modified by a $\delta\mathbf{x}_0$ amount, the corresponding trajectory is given by the corresponding flow map variation:

$$\delta\mathbf{x} = \phi(t; \mathbf{x}_0 + \delta\mathbf{x}_0) - \phi(t; \mathbf{x}_0) \tag{A3}$$

with respect to the reference orbit as indicated qualitatively by Figure A1 (taken from [15]).

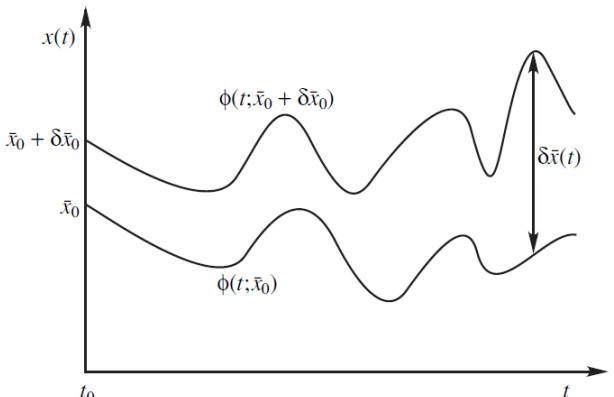

**Figure A1.** Nominal Trajectory, and Neighbouring Trajectory.

At this point, we can make "small" adjustments $\delta\mathbf{x}_0$ to the initial state $\mathbf{x}_0$ so that an adjusted trajectory will end up at the desired final state $\mathbf{x}_d$, near the original one $\mathbf{x}_1$. Without any correction, the new map is given by:

$$\mathbf{x}(t_1) = \phi(t_1, t_0; \mathbf{x}_0) = \mathbf{x}_d - \delta\mathbf{x}_0 \tag{A4}$$

The basic idea behind differential correction is then to make a small change at one end to get a desired point at the other end. Using an iterative procedure similar to a two-point boundary vale problem, the process leads to convergence, i.e.,

$$\phi(t_1, t_0; \mathbf{x}_0 + \Delta\mathbf{x}_0) = \mathbf{x}_d + \epsilon \tag{A5}$$

where $\Delta\mathbf{x}_0$ is the cumulative correction set, which yields the desired final state $\mathbf{x}_d$ within some tolerance $|\epsilon| \ll 1$. It is easy to see that a simple differential correction, or simple shooting, is nothing but a Newton's method of finding a root for the flow map $\phi$.

To find the correct adjustment, we can compute the sensitivity matrix of the changes in the final state $\mathbf{x}_1$ with respect to small changes in the initial state $\mathbf{x}_0$. The procedure is simply an approximation to the first variation of the map $\phi$ with respect to the desired target state $\mathbf{x}_d$, and given by Equation (A5).

$$\phi(t_1, t_0; \mathbf{x}_0 + \delta\mathbf{x}_0) = \phi(t_1, t_0; \mathbf{x}_0) + \frac{\partial\phi(t_1, t_0; \mathbf{x}_0)}{\partial\mathbf{x}}\delta\mathbf{x}_0 + H.O.T \tag{A6}$$

The linear approximation to the sensitivity matrix yields the state transition matrix $\Phi$ evolution, along the reference orbit thus:

$$\delta\mathbf{x}_1 = \Phi(t_1, t_0)\delta\mathbf{x}_0 \tag{A7}$$

and

$$\phi(t_1, t_0; \mathbf{x}_0 + \delta\mathbf{x}_0) = \mathbf{x}_d + H.O.T \tag{A8}$$

with the transition matrix satisfying Equation (A1), from standard system theory:

$$\dot{\Phi}(t, t_0) = \frac{\partial\mathbf{f}}{\partial\mathbf{x}}\Phi(t, t_0) \qquad with \qquad \Phi(t_0, t_0) = I_6 \tag{A9}$$

where $\frac{\partial\mathbf{f}}{\partial\mathbf{x}} = D\mathbf{f}(\mathbf{x})$ is the Jacobian matrix of the flow map.

From Equation (A1), the Jacobian matrix in Equation (A9) can be written as:

$$D\mathbf{f}(\mathbf{x}) = \begin{bmatrix} 0 & I_3 \\ -\tilde{U} & 2\Omega \end{bmatrix} \qquad with \qquad \Omega = \begin{bmatrix} 0 & 1 & 0 \\ -1 & 0 & 0 \\ 0 & 0 & 0 \end{bmatrix} \tag{A10}$$

and $\tilde{U}$ being the matrix of second partial derivatives of the effective potential with respect to x, y, and z:

$$\tilde{U} = \begin{bmatrix} \bar{U}_x x & .. & \bar{U}_x z \\ .. & .. & .. \\ \bar{U}_z x & .. & \bar{U}_z z \end{bmatrix}$$

The differential correction for finding a periodic solution requires the definition of both an initial state and a final state. Symmetries of halo orbits can be used to aid this step. Recall that halo orbits are symmetric about the xz plane (y = 0), and they intersect this plane perpendicularly ($\dot{x} = \dot{z} = 0$).

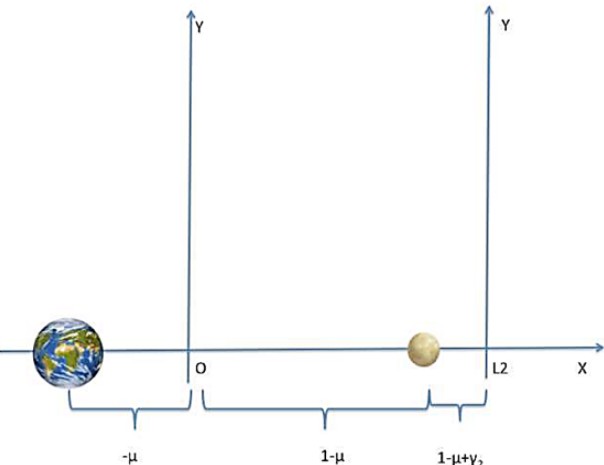

**Figure A2.** Reference Frame Transformation.

With reference to Figure A2, we can use Richardson's approximation solution [15] to find the initial state first guess:

$$\mathbf{x}_0 = [x_0, 0, z_0, 0, \dot{y}_0, 0]^T$$

The equations of motion are then integrated until the trajectory crosses the xz plane. To produce a periodic orbit, a perpendicular crossing is needed, and so the desired final (target) state vector must have the form:

$$\mathbf{x}_f = [x_f, 0, z_f, 0, \dot{y}_f, 0]^T$$

The first crossing of the xz plane occurs at a time equal to one half of the period of the orbit. It is likely, however, that actual values for $\dot{x}_f$ and $\dot{x}_f$ are not zero at the first crossing. The three non-zero initial conditions can be manipulated to drive these final velocities to zero. Recall that differential correction uses the state transition matrix to change initial conditions:

$$\delta\mathbf{x}_f = \Phi(t_f, t_0)\delta\mathbf{x}_0 \tag{A11}$$

The change $\delta\mathbf{x}_0$ in the initial state can be computed using Equation (A11) and the known change $\delta\mathbf{x}_f = \mathbf{x}_f^d - \mathbf{x}_f$ after the first iteration. Since this implies inverting a $2 \times 3$ state transition matrix, a more convenient approach is to constrain one of the initial state variables, for instance, $\delta z_0$ to zero. This makes $\Phi(t_f, t_0)$ a $2 \times 2$ square matrix and allows an easy computation of the other initial conditions $\delta x_0$ and $\delta\dot{y}_0$. At this point, we can start a second iteration with the new set of initial conditions $\mathbf{x}_0 + \delta\mathbf{x}_0$. The process continues until we achieve $\dot{x}_f = \dot{z}_f = 0$ within some desired bounds [6,28].

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
