# Peer review of "Phasing Maneuver Analysis from a Low Lunar Orbit to a Near Rectilinear Halo Orbit"

_aerospace, doi:10.3390/aerospace8030070_

Round 1

Reviewer 1 Report

ID: aerospace-1102661

Title: PHASING MANEUVER ANALYSIS FROM A LOW LUNAR ORBIT TO A NEAR RECTILINEAR HALO ORBIT

Authors: Giordana Bucchioni, Mario Innocenti

Journal: Aerospace

This paper investigates several analyses of phasing trajectories in terms of ∆V expenditure, time of flight, and position errors. The intensive simulation reveals the the total ∆V consumption, based on the scenario for Moon missions. The contents are interesting, and the structure of the paper is fine.

The reviewer has five concerns.

(1)

The content of this paper may seem to be a combination of previously established techniques. In this manuscript (introduction, abstract, and conclusion), new contribution is not clear. The authors should clarify which is novel here and what has already been proposed before, in the manuscript, so that readers could easily grasp the actual contribution of this paper. The clear statement of this paper’s contribution is mandatory.

(2)

Please check English spelling. The reviewer picked up five typos.

Lines 2-4: The working framework is the one proposed by the ESA’s Heracles mission in witch a passive target spacecraft is in a Near Rectilinear Halo Orbit and an active vehicle must reach that orbit to start a rendezvous procedure.

“witch” may be “which”.

Lines 7-8: The three approaches are compared in therms of ∆V consumption, accuracy and time of flight.

“therms” may be “the rms”.

Lines 65-67: The present paper focuses on the phasing trajectory defined within ESA's Heracles mission, in particular it presents three different methods for th phasing maneuver and compares them.

“th” may be “the”.

Lines 97-101: The x-axis points from the primary body to the secondary body, the z-axis is the axis of the angular velocity of the bodies and is therefore normal to the orbital plane of the two primaries and the y-axis supplements the coordinate system to be a right-hand system, it can be centerd on the secondary center or on the system's barycenter.

“centerd” may be “centered” or “centred”.

Lines 307-309: The figure shows also th loss of accuracy at the periselene, as the propagation time increases.

“th” may be “the”.

(3)

Table 2 is not easy to understand. In particular, the statement “> 50 Km below, > 86 Km behind” is unclear. Please add some explanations on “below” and “behind” clearly in the manuscript, or modify the table.

(4)

Table 3 shows “One T”. Is it just “1.0 T”?

(5)

Please unify the degree of a significant digit (a significant figure). Especially, the manuscript writes “In particular, a five period manifold propagation, and a phasing trajectory time of flight of 29d 5h 2min yield a position error within the requirements ad a ∆V = 678.401 m/s, as shown in Figure 16.” in lines 390-393. Here, the significant digit seems to be too much.

Author Response

  • the contributions have been stated more clearly in the introduction and conclusions. A reference has been added to provide additional information
  • The items in point 2 have been addressed and the English and spelling reviewed.
  • An appendix has been added describing the differential correction procedure.

Reviewer 2 Report

The paper studies the preliminary design of a phasing trajectory in a cislunar environment. Initial conditions to perform such transfer are studied through three approaches. Algorithms are proposed, numerical simulations are provided, and comparison of the three approaches is presented. The topic is interesting but the current version is not suitable for publication. A major revision is suggested. In the following detail my comments:

  1. English should be thoroughly refined. Many mistakes are found throughout the paper. For example:
    1. In Line 18, "For an ascending vehicle,a key...".  There should be a space between a comma/period and the next word "a". Similar errors occur throughout the paper.
    2. Many abbreviations are used before defined. For example, NRHO in Line 46, LLO in Line 51,  and so on.
    3. The first sentence in a new paragraph is suggested to be indented. 
    4. In Line 66, "... methods for th phasing maneuvers ...". An "e" is missing from "the".
  2. According to the definition of the coordinate system, the inertial frame adopts the earth centered inertial frame (ECI), which is slightly different from the lunar centered inertial frame.  Since the initial parking LLO is polar, did authors perform coordinate transformation to ECI when running numerical simulations?
  3. The boundary conditions in Table 2 are imprecise. For example, what is the exact orbit elements at departure? The location of departure may influence $\Delta v$ and TOF. At the insertion point, the condition ">50 km below" is given. So, is 100-km-below qualified? Is 1000-km-below qualified? is 1e5-km-below qualified?
  4. The right figure of Fig. 4 is drawn w.r.t LVLH. How is LVLH defined?
  5. In Page 6, authors claim that the initial conditions for target orbit propagation are normalized to earth-moon states (distance and speed). However, a normalized parameter is dimensionless, whereas the provided ICs have units. Please check this.
  6. Please provide brief mathematical description of differential correction, since this is employed as the main correction method in your research.
  7. Is Eq. (5) in vector form or in scaler form? If it is in scalar form, does this mean that all maneuvers are performed along the tangential direction? If so, then that may violates the Lambert's theory, because the Delta-v in Lambert's theory should be computed in a vector form.
  8. "End Point" and "Different" should not be capitalized in Fig. 6.
  9. In Line 283, the product sign "×" should not be replaced by the letter "x".
  10. The sentence below Eq. (10): "Where ...".  "where" should not be capitalized.
  11. In Section 5.3, authors claim that optimization procedure produces position errors. This is expectable, since a soft terminal constraint on  were imposed in Eq. (6). If a hard constraint were imposed, the result should have been different. As a result, I don't think this claim valid.
  12. The terminal condition is set as an 80-km stable manifold. Why 80 km? Does the offset distance influence  Delta-v or TOF significantly, or linearly?

Author Response

  • English language and typos have been addressed.
  • A list of abbreviations has been added to clarify the acronyms in the manuscript.
  • Point 2 has been answered by clarifying the use of the LLO orbit data.
  • Point 3 has been addressed in the Table.
  • Point 4 has been answered by adding the definition of LVLH frame.
  • Point 5 has been explained with reference to the normalized initial conditions.
  • Point 6 has been answered by adding teh appendix, which briefly desccribes the main ideas behinf differential correction.
  • Point 7, the DV expression is in vector form as explained in the revised manuscript.
  • Points 8-10 have been corrected.
  • Point 11. The use of soft constraints has been justified in lieu of the preliminary analysis, scope of the paper.
  • Point 12 was a requirement in the sample mission studied in the paper.

Round 2

Reviewer 2 Report

The revised version corrects errors and clarify some points in the previous version. However, I don't think the current version is good enough to publish. Some points are still not clear to me. A major revision is suggested again. In the following detail my comments:

  1. The modifications of contents are not highlighted, so it is very difficult to find the locations and differences of minor modifications. Highlight of modified contents is recommended in the revised version.
  2. I still cannot see the definition or transformation of the lunar inertial frame. 
  3. In my previous comment, I didn't concern the word description "below" or "behind" in Table 2. I was talking about ">50 km" and "> 86 km". What order of the offset is acceptable? For example, 100 km > 50 km; 10000 km> 50 km. Is 10000km acceptable as the insertion point? Please make the constraints on insertion point more precise if necessary.
  4. I don't think the definition of LVLH is appropriate. The current definition only applies to a perfect circular orbit. Given perturbations or an elliptic orbit, the definition will go wrong. Unfortunately, LLO is a highly perturbed environment. A better definition is suggested as follows: z points to the center of the moon, y points toward the angular momentum vector of the orbit, and x completes the triad. In the current definition, x may not be perpendicular to z in a perturbed or an elliptic orbit.
  5. Since DV is in vector form, bold font is suggested to distinguish from a scalar variable.
  6. I agree with the authors on the employment of soft constraint for preliminary analysis. However, position error is expectable in a soft constraint problem. Hence, authors should make this point clear. That is, position error may come from soft constraint, not the problem or algorithm itself. Therefore, I don't think the previous claim regarding position error valid.

Author Response

  1. the modifications are highlighted in red as requested by the reviewer.
  2. The transformation of the reference frames has been explained and two references added for confirmation. I apologize for the confusion, and I thank the reviewer for asking to clarify this point in the manuscript.
  3. The maximum distance sign was changed. In practice, the requirement provided to us was sort of soft in the range of the distance written in Table 2.
  4. The LVLH frame was detailed. In reality the axes are not striclty V-bar, H-bar, Z-bar as pointed out by the reviewer, but is is customary to address the components in that way.
  5. The equation is now in bold form.
  6. The optimization algorithm has been explained further. Our initial concern was to be able to compare relative error and fuel consumption at this preliminary phase.

We hope to have clarified the reviwer's concerns. Sincerely,

GIordana Bucchioni, Mario Innocenti

Round 3

Reviewer 2 Report

The revised version clarifies all my concerns, and I think this version is good enough to publish.